# Therapeutic Use of VR Serious Games in the Treatment of Negative Schizophrenia Symptoms: A Systematic Review

**DOI:** 10.3390/healthcare10081497

**Published:** 2022-08-09

**Authors:** Beatriz Miranda, Pedro Miguel Moreira, Luís Romero, Paula Alexandra Rego

**Affiliations:** 1ADiT-Lab—Applied Digital Transformation Laboratory, Instituto Politécnico de Viana do Castelo, 4900-347 Viana do Castelo, Portugal; 2LIACC—Laboratório de Inteligência Artificial e Ciência de Computadores, Universidade do Porto, 4200-465 Porto, Portugal

**Keywords:** virtual reality, game design, serious games, schizophrenia

## Abstract

Schizophrenia is a chronic brain disorder that affects 1 in every 300 people worldwide. This study intended to perform a systematic review to describe the state-of-the-art of interventions involving patients with negative symptoms of schizophrenia that use Virtual Reality (VR) games as a complement to therapy, and to analyze the key features of such games. Literature research was conducted in three databases, namely, the Institute of Electrical and Electronics Engineers (IEEE), Scopus, and PubMed, to identify relevant publications dated from 2010 to 2021. Of the initial 74 publications found, only 11 satisfied the eligibility requirements and were included in this study. The results were then organized and displayed in a flow diagram. Overall, the results from the studies suggest that the use of VR in therapies enables an increase in social skills and a decrease in anxiety symptoms. The use of such technology in therapy has proven to be effective, although it still lacks features to provide better long-term results.

## 1. Introduction

One of the key characteristics of schizophrenia has been recognized as psychosocial impairment, which includes a deficit in social cognition and social behavior [1]. Symptoms such as depression, cognition, and even social functions have all been found to be influenced by the use of VR environments [2]. Thus, by using an immersive VR environment, it is possible to create realistic situations which could possibly trigger negative symptoms in patients but, in a secure simulation, enable users to build their confidence and cope with their struggles better.

VR is an interesting tool that has started to be used in remediation therapies [1]. Although it is typically used as an exposure technique for specific phobias, VR has recently been applied, with encouraging results, to the study and treatment of schizophrenia [3]. It can offer the potential for a significant therapeutic benefit [4], since patients are more willing to enter challenging situations and experiment with alternative ways of responding [5]. Patients can put on a headset and be immersed in multiple situations ranging in difficulty, which may cause them to experience different levels of psychological distress [6].

Individuals, not only in their own life activities but also in treatment, may be hesitant with respect to face-to-face engagement [7]. The causes of this anxious withdrawal may be many: social anxiety, negative self-image, panic attacks, and a lack of confidence [5]. Studies have shown that psychosocial interventions in schizophrenia have positive effects on disease symptoms, treatment compliance, quality of life, social cognition, social functioning, and employment [1], generating positive perceptions and pleasant emotions. An improvement in relationships and peer motivation was reported [8] in group therapy sessions with digital games, where patients can play together. The power of VR interventions thus far has been most evident in the treatment of anxiety disorders [5]. Yet, some studies also noted an improvement in general psychopathology, negative symptomatology, and daily functioning [1].

The success of a rehabilitation program depends on several factors [4]—not only is exposure to a troubling situation key for clinical change, but actively testing out fears with the dropping of defense behaviors is as well [5]. The sense of embodiment seems to be one of the key factors in a VR application, especially when it focuses on the increase in empathy. The user must feel self-located inside the virtual environment, perceive feedback from the environment as if it was from his own sensations, and recognize himself as the cause of the actions [9].

VR therapy can be as effective, if not more effective, than treatments delivered in traditional formats [5]. Nonetheless, there is still an initial reluctance to use VR in working with patients with schizophrenia [1] due to the old misconceptions that games are only about having fun and cannot be used to develop practical skills. Consequently, the existing rehabilitation games lack the entertainment factor and should be updated to meet a greater number of key parameters, causing them to become more useful therapeutic instruments [4]. For instance, most VR therapy systems still require the presence of a therapist to administer the treatment, thereby restricting the treatment scalability [5]. Additionally, the effectiveness of computer games for rehabilitation could be enhanced by the incorporation of a social dimension [4,10]. Furthermore, to the best of our knowledge, there has been no study published in which games use collaboration or competitiveness as a significant part of the schizophrenia rehabilitation process.

To conclude, there is still a long way to go in developing a solution for VR game therapies. The aim of the present study is to conduct a systematic review of the interventions based on VR targeted at patients with negative symptoms of schizophrenia and also of the characteristics of such interventions. From such a review, an analysis of the presence or absence of key components of those interventions will be made.

## 2. Serious Games

Serious games are becoming a prominent study topic fueled by advancements in game production and computer graphics hardware, which are in turn fueled by the popularity of video games [4]. This has led to a view of games as educational technologies and, consequently, as having application beyond the realm of entertainment [11]. Although the term serious game still has a broad range of definitions, it is agreed that the main purpose of these games goes beyond mere entertainment [4,12,13,14]; they are designed to make players learn something and, if possible, have fun while doing it [14]. For the purpose of this article, serious games are defined as digital games that, through some form of simulation, allow users to build knowledge and learn new skills in a more engaging and fun way.

Games provide powerful and meaningful contexts for learning [11], as they can have positive impact on the development of different skills [14]. In addition, games have demonstrated that they help to enhance motivation in rehabilitation sessions, which is a big issue in treatment sessions due to the repetitive nature of the exercises [4]. 

The fact that everything takes place in a simulation allows players to develop specific skills [12,13,14] and experience situations that are difficult to reproduce in the real world for reasons regarding safety, cost, and time [14]. When combined with immersive VR, serious games can easily computationally simulate realistic virtual environments in order to recreate scenes and situations experienced in everyday life [13]. These VR based-methods can offer patients immersive experiences that are engaging and rewarding for them [4]. The growing diffusion of serious games also expands the advances of VR to society by bringing applications that motivate the assimilation and construction of new concepts in contexts similar to those in the real world [13].

## 3. Schizophrenia Symptomatology

There is recurring evidence substantiating the fact that negative symptoms are apart from positive symptoms. While positive symptoms are often the reason people get diagnosed with the illness, negative symptoms have been reported as among the most common early symptoms of schizophrenia, but they can occur at any phase [15].

In comparison, positive symptomatology is characterized as an excess or distortion of normal function, and it can include, for example, delusions and hallucinations. On the other hand, negative symptomatology is characterized by a decline or absence of normal behaviors and can include a decrease in motivation and interest [15].

In depth, according to Correll et al. [15], the negative domain consists of five symptoms: blunted affect (a diminution in facial expressions), alogia (a decrease in the number of words spoken), avolition (a reduction in goal-oriented activities), asociality (poor relationship management), and anhedonia (a decreased experience of pleasure). This domain can also be subdivided into primary and secondary negative symptoms. However, these symptoms can be hard to distinguish, but recognizing the differences between the two is crucial for clinical trial design, and researchers need to know how to differentiate them. Fundamentally, primary symptoms are the ones intrinsic to the disease, and they usually cannot be managed with the currently available treatments; on the other hand, secondary symptoms are the ones that occur as a result of positive symptoms, affective symptoms, medication side effects, and other related factors [15].

Finally, there is still a medical need for effective pharmacologic therapies to address negative symptoms. Thus, technologically advanced interventions aimed at addressing attitudes, behaviors, and psychosocial functioning can be beneficial when used in conjunction with the available treatments [15].

## 4. Materials and Methods

A systematic review of therapeutic studies was carried out using VR serious games targeted at schizophrenia negative symptomatology, as well as other relevant therapies. The systematic review strategy was conducted according to the Preferred Reporting Items for Systematic Reviews and Meta-analyses (PRISMA) guidelines [16].

### 4.1. Eligibilty Criteria

From the search results, only those that were written in English and included at least one of the keywords mentioned below were selected for revision. Preference was given to studies that focused on the development of 3D VR game therapies for the treatment of negative schizophrenia symptoms, although other suitable publications were also initially selected for revision. For instance, systematic reviews, duplicated studies, partially available studies, and the development of VR games for other rehabilitation purposes were initially revised but not included in the final revision.

Studies were excluded if they were:Literature reviews;Not specifically targeted at schizophrenia;Not focused on negative symptoms;Not involving game-based therapies or using a game developed previously by another study or entity.

### 4.2. Information Sources and Search Strategy

The three databases used to gather the dissertations, articles, and other publications of interest were IEEE, Scopus, and PubMed. Articles published from January 2010 to October 2021 were reviewed to obtain the most relevant and updated research. The 2010 start date was chosen since it represents a relevant milestone in terms of the development of virtual reality technologies. In addition to the technological improvements in the systems, the availability of high-speed internet connection, combined with the reduction in the costs of the devices, led to an increase in their use.

Then, the following keywords were combined to look for studies of interest in those databases: “Virtual Reality” or “VR” and “Schizophrenia” or “Negative symptomatology”.

### 4.3. Data Extraction

To assist in the selection of the articles, a specific Excel spreadsheet was prepared for extracting the data as well as for helping to identify the fulfilment of the inclusion and of exclusion criteria. The data extraction was completed by two authors (B.M. and P.A.R.). Where different decisions occurred, these were discussed, and the respective articles were reassessed.

### 4.4. Data Items

In the analysis performed on the selected articles, the following data items were extracted: sample, duration, therapy targets, interaction, immersion, scenery, adaptation, progress monitoring, feedback, portability, and automation.

### 4.5. Risk of Bias Assessment

Since the present review does not aim to study the effectiveness of interventions, an assessment of the risk of bias was not performed.

## 5. Results

The eleven studies that were selected were then analyzed according to what we considered to be key components in serious games. Some of the selected criteria followed the guidelines proposed in Rego et al. [17].

### 5.1. Study Selection

The steps shown in Figure 1 describe the selection process of the articles within the three databases used in the present study and the application of the inclusion and exclusion criteria, which led to 74 records being collected. After checking these 74 articles, in the Identification step, duplicates were excluded (n = 20). After the application of the exclusion criteria, in the Screening phase, only 35 articles were left. The remaining 19 records were then evaluated for eligibility, and 8 were excluded for the reasons stated in the Eligibility step (cf. Figure 1), resulting in 11 studies being included in the review. The studies were independently screened by two authors (B.M. and P.A.R.) without the use of any automation tools.

### 5.2. Classification Criteria

Since no specific criteria have yet been defined for the development of VR game therapies for schizophrenia, current available studies often overlook key parameters that can make a difference as to whether the game is going to be effective or not. Therefore, current studies on the subject were analyzed and compared with each other, following the proposed taxonomy on serious games by Rego et al. [17], to find out which ones implement the most parameters and how these are implemented. 

From the analysis that was carried out, eleven parameters related to the testing trials and to the game therapies were selected for comparison purposes. The classification criteria for those articles include: sample (the number of subjects tested), duration (the duration of game testing in terms of sessions, weeks, and/or months), therapy target (the target skills the game is supposed to improve), interaction (the interaction modality), immersion (whether it is immersive or not), scenery (essentially the game environment, but also the genre and storyline), adaptation (game difficulty settings), progress monitoring (whether or not the system can collect performance data during gameplay), feedback (the game’s response to user actions), portability (whether the game therapy can be administered outside of the health facility), and automation (whether or not the game is automated).

Table 1 illustrates the classification made for the selected VR therapies using the mentioned eleven parameters.

### 5.3. VR Game Interventions Description

#### 5.3.1. Social Skills Training VR (SST-VR) Role-Play 

Conversation skills training, assertiveness skills training, and emotion expression skills training are the three types of skills on the basis of which each one of the tasks is chosen to be completed by the patient in role-playing games representing everyday situations. This intervention sample was divided into two groups: one that used immersive VR in the sessions and the traditional group, which involved just the patient and the therapist role-playing. The main purpose of the study was to find advantages of the use of VR in social rehabilitation compared to the traditional forms of rehabilitation [18].

#### 5.3.2. VR Vocational Training System (VRVTS) 

The vocational role-playing takes place in a virtual boutique [19]. This scenario was chosen for the study due to the identification of a salesperson as someone who requires social skills to interact with customers and to handle conflicts as well as problem solving skills. In order to level up in their position, the participants had to complete three levels of difficulty, all with a final competence test.

The purpose of this study was to investigate the efficacy and effectiveness of VR for enhancing vocational traits. In this study, the sample was divided into three training groups: VR, therapist administered, and conventional. Despite only one of these groups being digital-based, the content and structure were similar.

#### 5.3.3. Virtual City 

Patients and their matched controls completed the same eight individual tasks in the virtual city [20]. Four tasks tested their ability to find different targets around the city, and the other four tested their ability to return to a specific location. This way, the study aimed at developing a VR game to function as a meaningful measure of cognition and complement cognitive tests during clinical trials for schizophrenia treatments.

#### 5.3.4. Soskitrain 

This VR game consists of seven activities based on chosen targeted social interactions—namely: criticism, social assertiveness, confrontation expression, heterosocial contact, interpersonal warmth, conflict/rejection by parents, interpersonal loss, and positive expression [3]. Thus, the game allows users to practice social interactions with virtual characters, encourages the learning of social skills, and provides feedback on the user’s actions. Additionally, in order to assess the performance of the patient, the game uses their committed errors, assertive behaviors, and time as the score.

This study aimed at reporting the results achieved with their VR game when it was used to complement schizophrenia interventions, hoping that it would improve the social cognition and performance of patients.

#### 5.3.5. Virtual Morris Water Maze and Carousel Maze 

This VR game intervention was designed to demonstrate the deficit of spatial cognition in schizophrenia [21]. It contemplates two levels, the Stable arena and the Rotating arena, designed to make users navigate towards several hidden goal positions placed on the floor of the enclosed arenas. The Stable arena had a virtual four-goals navigation task that included finding and remembering certain positions using three orientation cues. The Rotating arena was like the previous one but with two frames—one in which both the arena and the player rotate and another in which there is a static room that moves according to the position of the player.

#### 5.3.6. VR Vocational Rehabilitation Training Program (VR-VRTP) 

Two workplaces, believed to be more likely to employ people with schizophrenia, were developed: a convenience store and a supermarket [23]. Before entering a specific scenario, patients received training in presenting a good image, including training in greeting posture, smiling, and common verbal greetings. Each trial included practical situations commonly encountered when working on the chosen field of work. The game provided feedback as well as scores to give a sense of accomplishment as the users progressed. Likewise, the VR-based vocational game was built so that schizophrenia patients could learn skills to be applied in a real-life context.

#### 5.3.7. Serious Game to Improve Cognitive Functions in Schizophrenia 

A virtual city inspired by Paris was developed as the scenery for this serious game [24]. The existing separate places—in locations that could easily be found, such as a bank, a supermarket, a restaurant, a pharmacy, and a park, among others—could be used as visual landmarks to orient their navigation. The patients were expected to cooperate by sharing strategies, solving problems, planning actions, and using the 2D and the 3D map to reach their common goal, depending on the instruction they had to follow. This study implements a method that is expected to improve schizophrenia cognitive skills, especially memory, as well as planning and executive functioning tasks.

#### 5.3.8. Social VR Simulation 

The social interaction of this game consists of two tasks in a fixed order [22]. The first required that participants ask their virtual co-workers for help when handling a new program, while the second required the collection of money to buy a gift for the boss. It was necessary to interact with each one of the five coworkers twice—once to receive neutral/cooperative feedback and another time to receive negative/rejection feedback. Therefore, patients were matched with healthy controls to delineate psychological mechanisms for paranoid ideations and to test psychological interventions against paranoia.

#### 5.3.9. Virtual Supermarket Shopping Task (vSST) 

Each round was divided into two stages: Acquisition, where players were given a shopping list to memorize during a certain amount of time, and Recall, where they were required to randomly pick up items from the supermarket list [25]. There were five consecutive rounds, each with increasing difficulty, as the number of items to memorize and collect increased. Completing the tasks successfully meant not making Intrusion errors, by picking up the wrong item, nor Omission errors, by missing some of the items from the list. The development of this virtual supermarket game is aimed at testing and training the memory and executive functions of patients with schizophrenia.

#### 5.3.10. Multimodal Adaptive Social Intervention in VR (MASI-VR) 

Every game session required the participant to complete twelve social missions [26]. These were ordered by degree of difficulty (four easy, four medium, and four hard), which was determined by the number of conversational inquiries and responses required for mission completion. Once the missions started, the users were free to explore the virtual space and engage with the available avatars. Additionally, the players had to perform social interactions with no negative consequences; in other words, if they selected the wrong response, they would get feedback and the possibility to try again.

In this way, the study examined the feasibility and acceptability of improving social functioning in schizophrenia by making patients start conversations with strangers to ask for information.

#### 5.3.11. GameChangeVR Therapy 

The VR therapy begins in the coach’s room, where users meet their virtual automated coach [5]. All levels require the patients to carry out simple tasks such as ordering drinks, finding objects, and speaking to people. As the game progresses, the scenarios of the levels become busier and noisier, displaying anxiety-triggering elements such as Closed Circuit Television (CCTV) cameras, police officers, and people staring or standing in their way. In this way, it intends to test the fearful cognitions of patients while limiting their ability to use safety-seeking behaviors by challenging users to try to do something different. 

This study describes the process of developing an automated VR game therapy targeting the highly prevalent anxious avoidance of everyday situations by patients, with the use of a virtual coach who explains the psychological principles and guides the patient through the treatment.

### 5.4. Samples

The number of tested subjects encloses all patients who are a part of the target audience, and, thus, healthy controls are excluded.

The sample size varied between 7 [24] and 95 [19] participants, mostly schizophrenia patients from mental health facilities. Together with the subjects diagnosed with schizophrenia, several studies simultaneously performed trials with their matched healthy controls [19,20,22,23,24] to validate the efficiency of the experiment. The sample from Table 1 does not include any number of healthy controls.

### 5.5. Duration

The duration corresponds to the length of testing in terms of months and weeks and the number of therapy sessions during that period of time. With the exception of the single-trial studies, the duration of each therapy study varied between 2 weeks [22] and five months [19], with a frequency of once or twice a week and each session lasting from half an hour to one hour. Additionally, while some studies only conducted a single trial [20,21,25], others had a pre-assessment [3,19,22,23,24,26], in order to evaluate the diagnosis of the patient and to both choose the subjects that would benefit the most from the experiment and compare the trial outcomes, and/or a post-assessment [3,18,19,23,24,26], which was carried out a few months after the end of the initial therapy sessions in order to check the long-term results.

### 5.6. Therapy Targets

Social therapy interventions are often based on two critical human skills: behavioral social communication skills and/or cognitive social communication skills. While behavioral skills refer to verbal and non-verbal social behaviors such as conversation, facial expression recognition, attention, and manners, cognitive skills refer to executive functions such as flexible thinking, processing information, and emotional regulation. Some of the studies specifically described their study as targeted at behavioral or cognitive skills. Those that did not were considered behavioral or cognitive if they mostly focused on skills belonging to one or the other, as described above.

From the eleven articles that were reviewed, seven focused on cognitive skills and only four focused on behavioral skills. Although some studies took a more general approach, targeting several different parameters of those skills, others were very specific in their purpose. Some of those therapy targets were anxiety and avoidance [5], vocations [19,22], and paranoia [24].

### 5.7. Interaction

Interaction refers to the several types of interaction modalities used in systems, which allow users to interact with them. The most common ways of interacting with games are a mouse and keyboard, for PC-based games, and a joystick, for both PC-based and console-based games. Other than these commonly known controls, there are other less-known and emerging technologies. In the case of immersive systems, these require the use of a Head Mounted Display (HMD) and, usually, the use of VR joysticks, but they can also be developed to use regular joysticks or even hand recognition. More advanced types of interaction may also integrate voice, face, or motion recognition.

### 5.8. Immersion

With the increase in the popularity of VR, it is common to hear people referring to immersive VR as just VR, but in fact there are three categories: non-immersive VR, semi-immersive VR, and fully immersive VR. Since none of the reviewed articles include any semi-immersive systems or equipment, only the remaining two categories were taken into account.

When considering serious game interventions targeting mental issues, there is still no scientific evidence as to whether using immersive therapies is more beneficial. Both immersive and non-immersive VR programs can incorporate strategies for optimal learning via personalized exercises and rapid feedback [26]. On that note, it is natural that only about half of the studies were immersive, since using the immersivity factor can bring both advantages and disadvantages. While it can help them feel an increase in empathy, feel self-located in the virtual environment, and recognize themselves as the cause of actions, patients are also more likely to experience side effects, such as dizziness and nausea. Non-immersive systems are easier to set up at home, allowing for more intensive [26] and accessible treatments.

### 5.9. Scenery, Storyline, and Genre

There are plenty of serious games with abstract concepts focusing only on the development of a skill through the execution of certain exercises repeatedly. The problem with these types of games is that they often can get boring and repetitive. As the main reason to include a game in therapy is to keep the patients engaged while developing functional skills, it is beneficial if games have a narrative based on real-life events.

The scenery is the virtual locations where everything happens; it can be as simple as a room or as complex as a new world. The storyline is the narrative that the game follows; for example, the storyline can follow the life events of the main character. The genre is a categorization into which the game falls; for instance, if the game genre is simulation, it means it involves controlling the life of a single character or a group of characters. On the other hand, if the game genre is sports, it means it simulates sports with teams of real or generated players.

All the game therapies studied were targeted at either cognitive or behavioral skills and were meant to help patients develop skills to be used in daily life activities. As such, the predominant chosen game genre in the studies was Simulation, except for that in the study by Fajnerova et al. [21], who chose to develop a Role-Playing Game (RPG). Life Simulation games focus on replicating real-life activities in a simplified way so that players can learn a desired skill. It is very useful, especially when it simulates situations that are too expensive or too difficult to reproduce in real life.

The adopted scenery and storyline also followed the same concept of simulating real-life situations. In terms of scenery, public places such as shops were selected according to how often they appear in the lives of patients, the number of people present there, and the number of job opportunities, allowing developers to easily integrate several types of tasks. The storyline also included completing daily activities, such as interacting with people and completing job-related tasks and memory-related errands.

### 5.10. Adaptation

As suggested by Rego, Moreira, and Reis [17], the game system can have an adaptation mechanism defined before gameplay (Configuration), during gameplay (Adaptability), or in both of the previous moments. Or, it can have no adaptation (None). Pre-game play adaptation allows the therapist or the patient to set up the game attributes according to the personal needs of the patient before the game session starts. It could, for example, set specific parameters such as time or just choose the overall difficulty between easy, medium, or hard. During gameplay adaptation, the system can adjust its difficulty automatically as the game progresses based on the user’s performance. The common criteria used to measure the performance of the player are score and mistakes, among others.

Adaptation plays a key role in serious games, since it is not beneficial to make the patient consistently play something that is either too easy or too difficult. It is also a tool that the therapist can manipulate to increase the therapy efficiency in order to help patients develop specific skills. 

From the examples included in this review, only three studies [5,25,26] implemented the Adaptation mechanism, all in the form of pre-gameplay (Configuration). For instance, Plechatá et al. [25] explain that the vSST allows users to adjust the difficulty level by increasing the number of objects and the customization of several parameters. When Adaptation is not considered, some of the games [5,18,19,21,22,26] have several levels which differ in difficulty. 

### 5.11. Progress Monitoring

Games with this feature can collect data regarding patient performance. The type of data gathered depends a lot on the therapy target and the game itself. Yet, these games often collect basic information such as the time spent completing a task, accuracy/inaccuracy when performing the required actions, and level score. In relation to serious games, progress monitoring works as an excellent tool to help the therapist both analyze the session results and keep better track of the progress of the patient, as well as to help the self-motivation of the patient.

When evaluating the reviewed articles, games were considered to monitor the progress of the player if they stored and displayed data such as score or time. Thus, about half of those articles mention some form of progress monitoring, but in some of them, it is not clear whether the data is collected by the researchers or automatically by the game.

### 5.12. Game Feedback

Game feedback is provided to users to show them if they are progressing successfully through the game or not. Whether we think of a game or application, every action needs to result in some form of feedback to help guide users through the steps they need to perform to achieve a certain goal.

Regarding the types of feedback, those can be based on [17]:**System interface**: when it is presented in the form of cause–effect—the user does something and the system reacts accordingly;**System-controlled**: when the game measures the performance—the user receives hints that guide him through the game;**Therapist-controlled**: when it is not automatically generated by the system—the therapist is the one who controls the feedback, so it is totally personalized to fit the therapy goals;**Mixed**: when the system uses two or more of the above types of feedback.

The most common type was system interface [18,19,22,23,26], but there were also references to system-controlled [5,26] and therapist-controlled [18,19] feedback. In addition, feedback can be provided in the form of visual, auditory, and haptic feedback or a mixture of these. Most of the games mentioned had either visual feedback, auditory feedback, or both representations, but none mentioned any form of haptic feedback.

### 5.13. Therapy Portability

Game portability evaluates whether a game therapy can take place only at the hospital, clinic, or other type of health facility where the treatment is taking place, or if it can also be administered at the patient’s home. In case it can be administered at the patient’s home, but there is a need to have the presence of the therapist throughout the therapy session in order to guide and assess the patient, it is considered ‘Home-Assisted’. It is considered ‘Home’, if it can be used at the patient’s home and there is some in-game therapist that automates the process, otherwise it is categorized as ‘Clinic’. This evaluation can also depend on the necessary equipment and its setup; for instance, a non-immersive game requires less equipment and is easier to set up, making it more accessible to be used at home.

In the reviewed game therapies, only two made reference to their portability. Among those, gameChange [5] was developed with the intention to make its therapy completely portable. Adery et al. [26] note that non-immersive systems such as MASI-VR can easily be used at home, so the game could be used as an adjunct social intervention together with more intensive treatments requiring skilled therapists and in-person attendance, but they do not specifically classify it as portable.

### 5.14. Automation

Currently, there are still only a few studies that rely on the use of automated therapy interventions. Such a feature enables treatments to be delivered automatically by a virtual coach without the need for a trained therapist to be present [5]. Automated psychological therapy has the potential to produce large clinical benefits and to increase treatment provisions for mental health disorders at a large scale and low cost.

From the reviewed articles, those considered automated were the ones that implemented a virtual therapist whose purpose was to guide users throughout the game. Only one characterizes its intervention as automated by including a virtual coach [5], but it was not the only one that used some form of automation to enhance the game experience. Other studies implemented what they call a personal assistant [22] or game narrator [26], who gives instructions, tips, and/or feedback to help guide the player through the game.

Additionally, a more efficient strategy for automating game therapy would use an Artificial Intelligence (AI) system. With the help of AI, the virtual therapist would be able to better understand the users’ needs and help them accordingly. Yet, none of the studies make any reference to the use of AI.

## 6. Discussion

To sum up the results, the eleven studies had a sample that ranged between 7 [24] and 95 [19], and the duration of tests consisted of multiple sessions lasting 30 to 60 min, once or twice a week, and from two weeks [22] to five months [19]. The therapy target of most of the studies involved cognitive behavioral skills. As for the interaction equipment, the games that were immersive used an HMD [3,5,18,22], and those that were not used either a mouse and a keyboard [19,23,26] or a joystick [19,20,21,24]. The games were all based on the simulation of real-life activities such as having social interactions with strangers on the street, looking for stores in a small city, and performing work-related tasks. As for the adaptation, only three games [5,25,26] had integrated pre-gameplay adaptation. About half of the games monitored the progress of the player. The most common type of feedback was system interface [18,19,22,23,26]. Only two studies [5,26] made any reference to having implemented game portability, and only three studies [5,22,26] implemented an automated game.

In terms of outcomes, these were mostly positive, and most of the hypotheses were confirmed. In terms of results, these varied and were dependent on what each study was aiming at achieving with their therapy system. Overall, all of the studies were able to reduce some negative symptoms by increasing the lacking skills [3,26]. In terms of cognitive skills, they registered an increase in memory, attention, confidence, and the recognition of their own emotions and difficulties, as well as a reduction in anxiety and withdrawal [3,19,23,24]. In terms of behavioral skills, they registered an increase in organization, work performance, and assertive behaviors, as well as an increase in social and conversational skills and a decrease in social avoidance [3,18,19,24].

Still, some of them did not register enough improvements or did not register any improvement at all in terms of skills such as vocal skills (most likely due to the lack of real-life speaking involved in the game), nonverbal social skills (because players did not need to use facial expressions or gestures while playing), and time management (due to a deficient performance in time-based actions). 

In particular, three articles decided to prove the cognitive deficit caused by negative symptoms in people with schizophrenia. Plechatá et al. [25] was able to demonstrate that all the patients tested were impaired in all the assessed cognitive domains, confirming their deficit in cognitive abilities. Hesse et al. [22] verified that it is possible to trigger stress and paranoid ideation with the use of VR, which also proves the need for an appropriate design target for this specific group of people. They also verified that, when the stress levels were too high, the symptoms would last must longer. Zawadziki et al. [20] demonstrated that individuals with schizophrenia have significantly more difficulties in navigation-related tasks.

Some studies also assessed patients’ feedback regarding the game therapy. When compared with traditional therapy sessions with no use of any kind of VR, patients rated the training programs as more interesting, useful, engaging, and motivational [18,19,26]. Overall, patients rated the systems as a stress-free environment, easy to use, helpful, and immersive [5,19].

Additionally, five papers revealed that they had plans to perform more trials to test the systems in a bigger sample and to test some updates that were added based on the results found [5,20,24,25,26].

Regarding other systematic literature reviews, it is safe to say that they took a different approach from ours. For instance, Sotos et al. [1] studied VR for psychosocial remediation in schizophrenia and described the interventions based on the sample, duration, type of study, immersivity, therapy targets, and evaluation instruments. Bisso et al. [27] studied VR applications in the schizophrenia spectrum and described the interventions based on the therapy target, country of development, study design, sample by number and by patient’s diagnosis, Diagnostic and Statistical Manual (DSM), type of VR therapy, duration by number of sessions, time of each session and treatment time, and principal outcome assessment. Pavlidou and Walther [28] studied VR as a tool in the rehabilitation of movement abnormalities in schizophrenia and did not perform a characterization of the available interventions in terms of study trials or game design.

The goal of our study was to characterize the game interventions based on the proposed taxonomy for serious games of Rego et al. [17]. Thus, when it comes to other similar reviews, this study adds further characterizations of the game design and technological aspects of the available interventions, which are not present in the other reviews.

Lastly, it is important to consider several limitations when interpreting the results. Studies were not included if they were not published in the English language, which may influence the outcome of our results and the dissemination of the output data. Additionally, most of characteristics used to characterize the analyzed studies were based on the model proposed by Rego et al. [17].

## 7. Conclusions

Currently, VR game therapies seem unable to fully replace traditional face-to-face therapies. However, when combined with regular therapy, VR systems can help to achieve much longer-lasting results and keep patients motivated. The studies reviewed showed improvement in both cognitive and behavioral social skills.

Despite all the recent developments in VR systems, there are still some limitations. For instance, when virtual characters are used to interact with the patient, it is necessary to program their animation in advance, which may restrain their behavior to unrealistic actions. Moreover, patients are always dependent on the required equipment, which can be as simple as a standalone VR headset or as complex as a desktop computer plus two joysticks, a VR headset, and a bunch of cables. When it comes to immersive systems, the required HMD can become a hassle, since it covers about half of the face of the patient, which can prevent a proper evaluation of facial expressions, and when there is a need for cables, these can interfere with the movement and posture of the patient and, consequently, its evaluation. Still, whether the game is immersive or has realistic graphics seems to have little impact on the performance of the patients, as long as they can stay engaged in the activities. Difficulty level adaptation is still either presented in the form of configuration or not presented at all; it would be more helpful if the game was able to adapt to the performance of the player, promoting an efficient learning experience.

Finally, from the obtained results, we were able to identify some interesting research opportunities. For instance, no study referenced the use of AI; however, this fast-growing technology has a lot of potential to be used to automate VR systems. In addition, it has been shown that group therapy can be beneficial for patients with anti-social disorders. Furthermore, the development of multiplayer serious games could provide a much better user experience without the need to make use of unnatural character interactions. Instead, they would interact with actual people, which can potentially help them to obtain the expected results faster. The problem of group therapy remains in the form of finding a way to conciliate patients with different illness severities who progress at different velocities. Finally, when verifying whether the selected game therapies had implemented our selected key features, we realized that, frequently, those were not present.

## Figures and Tables

**Figure 1 healthcare-10-01497-f001:**
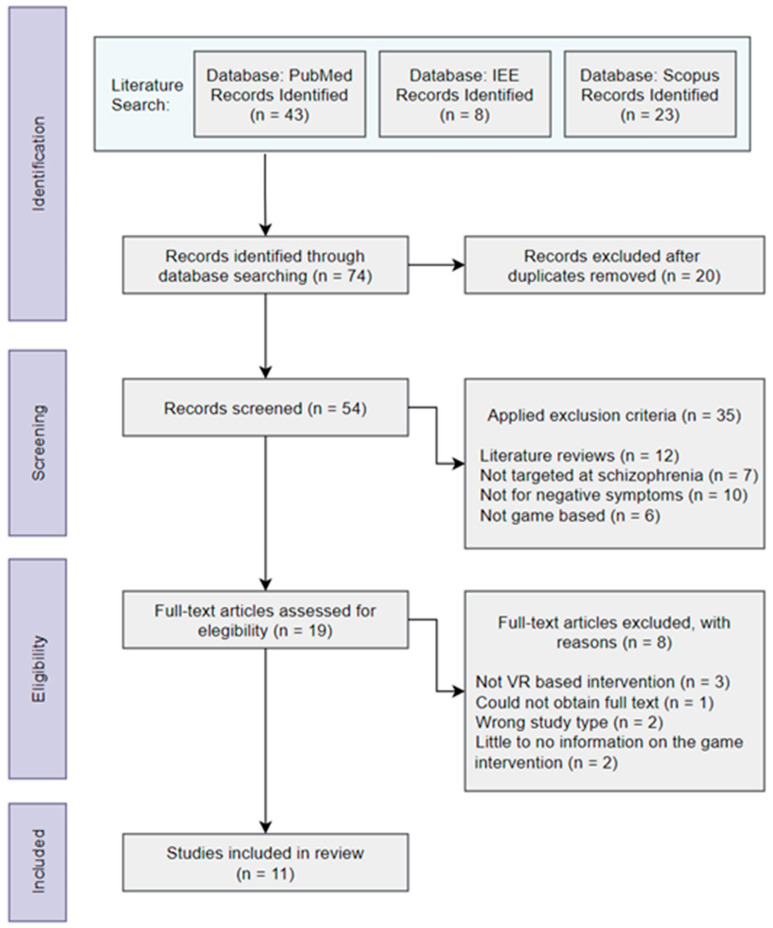
Steps followed to collect and select articles for revision.

**Table 1 healthcare-10-01497-t001:** The selected VR therapies classification.

Game Therapy Name	Authors, Year	Sample	Duration	Therapy Targets	Interaction	Immersion	Scenery	Adaptation	Progress Monitoring	Feedback	Portability	Automation
SST-VR ^1^ Role-play	Park et al., 2011 [18]	91 inpatients from a mental health hospital	10 sessions, twice a week over 5 weeks	Behavioral	HMD, joystick, voice, and motion tracker	Yes	House, shop, and street	--	--	Mixed	--	--
VRVTS ^2^	Tsang et al., 2013 [19]	95 inpatients who attended a vocational rehabilitation program	10 sessions, once a week over 5 months, lasting for 30 min	Cognitive	Keyboard, mouse, and joystick	No	Shop	--	--	Mixed	--	Yes
Virtual City	Zawadzki et al., 2013 [20]	33 patients from a mental health center	1 session	Cognitive	Joystick	No	City	--	Yes	--	--	--
Soskitrain	Calafell et al., 2014 [3]	12 outpatients from an adult mental health service	16 sessions, twice a week over 8 weeks	Behavioral	HMD, voice, and facial recognition	Yes	Shop and bar	--	Yes	--	--	--
Virtual Morris Water Maze and Carousel Maze	Fajnerova et al., 2015 [21]	29 first episode schizophrenia patients	1 session	Cognitive	Joystick	No	Maze	--	--	--	--	--
Social VR Simulation	Hesse et al., 2016 [22]	26 patients with psychotic disorders	2 sessions in 2 weeks	Cognitive	HMD, joystick, and voice	Yes	Office	--	--	System interface	--	Yes
VR-VRTP ^3^	Sohn et al., 2016 [23]	11 outpatients from a mental health center	8 sessions, once a week over 8 weeks, lasting for 60 min	Behavioral	Keyboard, mouse, and voice	No	Convenience store and supermarket	--	Yes	System interface	--	Yes
Serious Game to Improve Cognitive Functions in Schizophrenia	Amado et al., 2016 [24]	7 patients with schizophrenia institutionalized for many years	12 sessions, once a week over 3 months, lasting for 90 min	Cognitive	Joystick	No	City	--	--	--	--	--
vSST ^4^	Plechatá et al., 2017 [25]	26 subjects without any neurological or psychiatric diagnosis	1 session	Cognitive	--	--	Supermarket	Configuration	Yes	--	--	--
MASI-VR ^5^	Adery et al., 2019 [26]	17 outpatients from day facilities	10 sessions, twice a week over 5 weeks	Behavioral	Keyboard and mouse	No	Shop and bus stop	Configuration	Yes	Mixed	Home-assisted	Yes
gameChange	Lambe et al., 2020 [5]	11 patients with a lived experience of psychosis	6 sessions, lasting for 30 min	Cognitive	HMD	Yes	Shop, street, bar, bus, and doctor’s office	Configuration	--	System-controlled	Home	Yes

^1^ Social Skills Training; ^2^ VR Vocational Training System; ^3^ VR Vocational Rehabilitation Training Program; ^4^ Virtual Supermarket Shopping Task; ^5^ Multimodal Adaptive Social Intervention in VR.

## Data Availability

Not applicable.

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
