# Peer review of "Therapeutic Use of VR Serious Games in the Treatment of Negative Schizophrenia Symptoms: A Systematic Review"

_healthcare, 2022, doi:10.3390/healthcare10081497_

Round 1

Reviewer 1 Report

Authors congratulation for this Review and respective manuscript. The thematic of the paper – Therapeutic use of VR Serious Games in teh treatment of Schizophrenia Negative Symptoms -  is an important topic that deserves to be researched. This Review has originality and novelty.

However, some aspects need to be reformulated and improved.

Abstract:

Suggestion: indicate the population (patient with schizophrenia) in the summary, for example in line 12;

Suggestion: put the objective (/question) of this review.

the authors mention that the research had a time limit from 2010 to 2021 - why only include articles after 2010? there must be a reason, first article in the area is in 2010?

Keywords: I doubt it makes sense to use VR as a keyword, when you already have virtual realiaty ( confirm, please) 

Introduction:

Line 55 to 56 – remove the sentence as it is already on line 36;

Line 70 to 71 – Suggestion: the population is not schizophrenia, but people with negative symptoms of schizophrenia.

Through the analysis of the paper, interventions based on VR targeted at person with negative symptoms of schizophrenia were mapped. But not only the interventions were mapped, but also the characteristics of these interventions such as interaction level, immersion …;

This is not clear in the objective; therefore, suggestion to improve the objective;

Suggestion: put the objective at the end of the introduction/at the beginning of the method.

Method

the date of the search is an important data for replicability, as well as the development of future reviews on this topic; 

As already mentioned in the abstract, why only articles published after 2010?

The concept map used in the research is limited. Therefore, it is suggested mentinated it as a limitation, at the end of the discussion or in the conclusion.

The Data collection process remains to be explained – “specify the methods used to collect data from reports, including how many reviewers collected data from each report, whether they worked independently, any processes for obtaining or confirming data from study investigators, and if applicable, details of automation tools used in the process.”

Line 157-158: The text presented does not seem to respond to the Risk of Bias Assessment.

Regarding the text presented in line 158-159, given that the question/objective of this review does not focus on the effect of the intervention, the meta-analysis no longer makes sense, being even impossible to be conceived, given the objective that this review presents. ; this is not a problem, because the authors do not want to find the effectiveness of the intervention, but to map interventions. So the text on line 158-159 seems to make no sense.

Results:

It is suggested to rethink the presentation of finding in the results. The results should only include the findings from the included articles (and not from other articles that were not included);

Another suggestion: the finding  presented can be organized in a more logical order, e.g. as shown in table 1.

We also suggest the inclusion of the Discussion section, where authors can intersect the results found and synthesized with other findings from other studies and discuss/compare their own findings.

The authors' conclusion. The use of references is not recommended. The limitations of the review should be included (here or in the discussion).

Also due to some methodological weaknesses, it can be considered that the work scientifically has some weaknesses.

Regarding the Table, congratulations are given to the authors, as the manuscript is enriched:

However, it is suggested that:

- associated with the author and year, put the corresponding numerical reference;

- it is not necessary to use the letter in italic format;

- and at the bottom of the table (to make reading easier, put captions for the acronyms used)

Reviewer 2 Report

The manuscript by Miranda et al. is a very interesting and important study that clearly demonstrate the need for further VR development in Schizophrenia research.  Overall, I find the structure of the entire study and the presentation of the reviewed publications very valuable.  In particular, the  main findings have been adequately presented. 

Although I really appreciate the authors' work, and before recommending acceptance, I must also request authors addressing the following improvements.

The authors did not include literature reviews in the manuscript, which is acceptable for the structure of the Results section, but they should include such studies in their Conclusions section or in their Results section if they were to change the Results section to the Results and Discussion section.

Proposed paper 1: Immersive Virtual Reality Applications in Schizophrenia Spectrum Therapy: A Systematic Review

Proposed paper 2: Using Virtual Reality as a Tool in the Rehabilitation of Movement Abnormalities in Schizophrenia

Please address the similarities and differences of your findings with the current review literature and clarify the need for your review.

Minor:

Please add an empty space between the words and the References for the following lines:

L 48: various factors[4]

L 54: actions[9].

L 313: months[19]

L 488: [24]verified that

Round 2

Reviewer 1 Report

I congratulate the authors for the improvement process carried out.

Author Response

Dear reviewer

We thank you for your suggestions.

We have added a new revised version with an update to the english language and style.

Best regards,

Beatriz Miranda, Pedro Miguel Moreira, Luís Romero, Paula Alexandra Rego

Reviewer 2 Report

I am very glad to see that the authors replied to all of my comments and  improved their manuscript. I accept it in present form.

Congratulations!

Author Response

(The authors gave the same response as above.)
